# Characterization of Endofungal Bacteria and Their Role in the Ectomycorrhizal Fungus *Helvella bachu*

**DOI:** 10.3390/jof10120889

**Published:** 2024-12-23

**Authors:** Caihong Wei, Mengqian Liu, Guoliang Meng, Miao Wang, Xin Zhou, Jianping Xu, Jianwei Hu, Lili Zhang, Caihong Dong

**Affiliations:** 1State Key Laboratory of Mycology, Institute of Microbiology, Chinese Academy of Sciences, Beijing 100101, China; 18399410069@163.com (C.W.); liumengqian19@mails.ucas.ac.cn (M.L.); menggl@im.ac.cn (G.M.); mwang2136@gmail.com (M.W.); zhouxin@im.ac.cn (X.Z.); 2College of Life Science and Technology, Tarim University, Alar 843300, China; skyhjw@taru.edu.cn; 3Xinjiang Production and Construction Corps Key Laboratory of Protection and Utilization of Biological Resources in Tarim Basin, Alar 843300, China; 4Department of Biology, Institute of Infectious Diseases Research, McMaster University, Hamilton, ON L8S 4K1, Canada; jpxu@mcmaster.ca

**Keywords:** endofungal bacteria, full-length 16S rRNA sequencing, *Stenotrophomonas maltophilia*, *Variovorax paradoxus*, metabolism

## Abstract

*Helvella bachu*, an ectomycorrhizal fungus, forms a symbiotic relationship with *Populus euphratica*, a rare and endangered species crucial to desert riparian ecosystems. In this study, endofungal bacteria (EFBs) within the fruiting bodies of *H. bachu* were confirmed by a polyphasic approach, including genomic sequencing, real-time quantitative PCR targeting the 16S rRNA gene, full-length and next-generation sequencing (NGS) of the 16S rRNA gene, and culture methods. The genera *Stenotrophomonas*, *Variovorax*, *Acidovorax*, and *Pedobacter* were abundant in the EFBs of fruiting bodies associated with three *Populus* hosts and were consistently present across different developmental stages. Notably, *S. maltophilia* and *V. paradoxus* were detected in high abundance, as revealed by full-length 16S rRNA sequencing, with *S. maltophilia* also isolated by culture methods. KO-pathway analysis indicated that pathways related to primary, secondary, and energy metabolism were predominantly enriched, suggesting these bacteria may promote *H. bachu* growth by producing essential compounds, including sugars, proteins, and vitamins, and secondary metabolites. This study confirmed the presence of EFBs in *H. bachu* and provided the first comprehensive overview of their structure, functional potential, and dynamic changes throughout fruiting body maturation, offering valuable insights for advancing the artificial domestication of this species.

## 1. Introduction

*Helvella bachu*, commonly known as the Bachu mushroom, is a delectable edible fungus found in the Xinjiang Uygur Autonomous Region of China [1], as well as in Pakistan [2] and Iraq [3]. Research indicates that this mushroom can effectively enhance leukocyte phagocytosis, lymphocyte transformation rates, and antibody titers [4,5], while also exhibiting antioxidant activity and a strong inhibitory effect on HepG2 cells [6]. In recent years, it has attracted considerable attention due to its notable economic, nutritional, and medicinal values. However, the annual yield of this mushroom is limited, and fruiting bodies cannot be cultivated, as pure cultures have yet to be successfully established despite years of efforts.

Our previous studies have demonstrated that *H. bachu* is an ectomycorrhizal fungus. To date, *H. bachu* has been shown to establish a symbiotic relationship with various poplar species, including *Populus euphratica*, *Populus pruinosa*, and *Populus alba* var. *pyramidalis* [7] in desert riparian ecosystems. This association plays a crucial role in desert afforestation, but complicates efforts to obtain pure cultures of *H. bachu* mycelium. Additionally, during the isolation process, bacterial growth is consistently observed around tissue blocks, while mycelial germination remains absent. This suggests the possible presence of endofungal bacteria (EFBs) within the fruiting bodies of *H. bachu*.

EFBs are bacterial symbionts that reside within fungal hyphae and spores [8]. The presence of EFBs was initially reported by Mosse (1970) through electron microscopy [9], revealing their presence in the cytoplasm of *Endogone* spores. These bacteria have been identified in a variety of fungal species, including both mycorrhizal and saprophytic fungi, and have had a significant influence on the evolution of life, continuing to shape the ecology of a wide range of species [8,10].

In recent decades, the EFBs associated with mushrooms have been extensively researched. These EFBs exhibited remarkable diversity, with a wide range of taxa identified across various mushroom species. A total of sixty-six EFB isolates were obtained from wild-growing mushrooms [11]. Furthermore, a greater abundance of EFBs has been reported in ectomycorrhizal mushrooms. For instance, *Proteobacteria* was found to be the most abundant phylum of EFB in the fruiting body of *Tylopilus neofelleus*, accounting for 94.2%, with *Rhizobiales* and *Burkholderiales* also being predominant in this fruiting body [12]. In the dissected fruiting body of *Tricholoma matsutake*, *Micrococcales*, *Bacillales*, *Caulobacter*, and *Sphingomonas* emerged as the principal bacterial taxa across its various compartments [13]. Additionally, members of the *Allorhizobium–Neorhizobium–Pararhizobium–Rhizobium* complex were found to be most abundant in the fruiting bodies of *Cantharellus cibarius* [14]. Twelve potential endophytic nitrogen-fixing bacteria, including *Paenarthrobacter*, *Exiguobacterium*, and *Paenibacillus,* have been isolated from *Floccularia luteovirens* [15].

The ecological roles of EFBs in mushrooms have been recognized as both diverse and significant, contributing to various aspects of fungal health and productivity. EFBs could serve as source of bioenergy for fungal hosts, producing essential compounds such as sugars, proteins, vitamins, and auxins that promote hyphal growth and spore germination [16,17,18]. They facilitated nutrient transport, particularly of phosphates, to mycorrhizal structures, thereby enhancing respiration and adaptability to varying environmental conditions [19,20,21]. Certain EFB species, especially those from the *Pseudomonas* genus, have been identified as potential biological control agents against diseases such as brown blotch and internal stipe necrosis in *Agaricus bisporus* [11]. Moreover, EFBs associated with *Chanterelles* have been shown to positively influence fruiting body formation [22], while EFBs in ascocarp-associated bacteria may play a role in the development, maturation, and even the aroma of the Black truffle, *Tuber melanosporum* [23]. Overall, the intricate interactions between mushrooms and EFBs have provided valuable insights into fungal biology and presented opportunities for improving mushroom cultivation practices.

Understanding the microbiota is essential for the successful domestication of *H. bachu,* a valuable ectomycorrhizal fungus associated with *P. euphratica*, an important species in desert riparian ecosystems [24]. In the present study, EFBs were identified in the fruiting bodies of *H. bachu* during genome sequencing analysis. Their presence was confirmed through real-time quantitative PCR targeting the 16S rRNA gene. The bacterial abundance and community composition of EFBs associated with three types of *Populus* plants were then characterized using full-length 16S rRNA gene sequencing, along with predictions of their functional roles. Biomarkers specific to EFBs linked to different host plants were also identified. Additionally, EFBs from *H. bachu* at different developmental stages were analyzed through amplicon sequencing. Finally, EFBs isolated from *H. bachu* fruiting bodies using culture methods showed partial consistency with the high-throughput sequencing results. This study confirmed the presence of EFBs in *H. bachu* and provided the first comprehensive overview of their structure, functional potential, and dynamic changes throughout the maturation of *H. bachu* fruiting bodies.

## 2. Materials and Methods

### 2.1. Genome Sequencing and Discovery of EFBs in the Fruiting Body of Helvella bachu

The fruiting bodies of *H. bachu* (Figure 1A) were collected at Tarim University in Alar City, Xinjiang Uygur Autonomous Region, China. Following the removal of debris and rinsing with sterile water, tissue blocks were carefully excised from the inner base of the *H. bachu* fruiting body (Figure 1B).

High-quality genomic DNA was extracted following the manufacturer’s protocol, using the QIAGEN Genomic Kit (Qiagen, Düsseldorf, Germany). Genome sequencing was performed using the MGISEQ-T7 and PacBio Sequel II platforms, provided by Nextomics Biosciences Co., Ltd. (Wuhan, China). A next-generation sequencing (NGS) library with an average insert size of 350 base pairs (bp) was first prepared following the MGISEQ standard protocol and sequenced on the MGISEQ-T7 platform. Additionally, a 20 kb library was constructed according to PacBio’s standard methodology, with its quality assessed on the Agilent 2100 Bioanalyzer (Agilent Technologies, Santa Clara, CA, USA) prior to sequencing on the PacBio Sequel II platform (Pacific Biosciences, Menlo Park, CA, USA).

NGS reads were initially filtered using Fastp v0.20.0 [25] with default parameters. The filtered NGS reads were subsequently analyzed with Kraken v2.1.3 [26] to assign taxonomic labels to short DNA sequences (Figure 1C). Following this, a BLAST search (v2.7.1) [27] was conducted using the 16S ribosomal RNA gene sequence of *Escherichia coli* (MN900682.1) against the original sequenced data, with an e-value threshold set at 10^−5^. These reads were then subjected to an online BLAST analysis on the NCBI bacterial database (https://blast.ncbi.nlm.nih.gov/Blast.cgi, accessed on 10 November 2023) using the same e-value threshold to identify the taxonomic classification of these bacteria (Figure 1C).

### 2.2. Analysis of EFBs in Helvella bachu Fruiting Bodies Using Absolute Quantitative PCR

The EFBs of *H. bachu* was quantified by absolute quantitative real-time PCR, following previously reported methods [13]. DNA was extracted from the clean tissues of the *H. bachu* fruiting body using a FastDNA SPIN Kit for Soil (MP Biomedicals, Inc., Santa Ana, CA, USA) in accordance with the manufacturer’s instructions. Primers 338F and 518R (338F: ACTCCTACGGGAGGCAGCAG; 518R: ATTACCGCGCGGCTGCTGG) were selected to amplify the V3–V4 region of the 16S rRNA gene sequence [28].

Quantitative PCR (qPCR) was performed in a 10 μL reaction volume using a Bio-Rad CFX Connect Real-Time PCR Detection System (Bio-Rad Laboratories Inc., Hercules, CA, USA). The reaction mixture consisted of 5 μL of Vazyme SYBR Green Master Mix (Vazyme International LLC., Nanjing, China), 0.2 μL of each forward and reverse primer, and 1 μL of DNA template. A plasmid containing a 16S rRNA gene fragment compatible with the bacterial primer pair 338F/518R served as the quantification standard.

The amplified products from *H. bachu* tissue DNA, generated using the universal 16S rDNA primers (338F/518R), were purified via gel electrophoresis, cloned into the pLB vector, and positive clones were selected for plasmid extraction and concentration measurement. The initial plasmid DNA concentration was 4.7 × 10^12^ copies/μL, which was serially diluted tenfold to quantify bacterial 16S rRNA gene copies. Utilizing the standard curve construction method, qPCR was performed on DNA extracted from *H. bachu* tissue samples, with three parallel sets of reactions established to detect Cq values.

The reaction conditions included an initial incubation at 95 °C for 15 s, followed by 40 cycles at 60 °C for 34 s, during which fluorescence signal detection occurred at 60 °C. Melting curves were evaluated for each reaction to confirm the specificity of amplification. All reactions were conducted in technical triplicates, with nuclease-free water samples serving as negative controls across all plates.

### 2.3. Full-Length 16S rRNA Gene Sequencing by PacBio

The mature fruiting bodies of *H. bachu* associated with three different host plants, including *P. euphratica*, *P. pruinosa*, and *P. alba* var. *pyramidalis*, were harvested separately (Table 1). Clean tissue blocks weighing about 0.5 g were taken from the inner base of each sample, with three replicates per sample, and used for the full-length 16S rRNA gene sequencing by PacBio platform at Wuhan Grandomics Biosciences Co., Ltd. (Wuhan, Hubei, China).

Genomic DNA was extracted using the CTAB method with a Grandomics Genomic Kit (Grandomic Biosciences Co., Ltd., Wuhan, China), in accordance with the manufacturer’s standard operating procedure. DNA purity was assessed using a NanoDrop™ One UV-Vis spectrophotometer (Thermo Fisher Scientific, Waltham, MA, USA), yielding an OD260/280 ratio between 1.8 and 2.0 and an OD260/230 ratio between 2.0 and 2.2. The DNA concentration was further quantified using a Qubit^®^ 4.0 Fluorometer (Invitrogen, Carlsbad, CA, USA).

For 16S rRNA sequencing, the mass-seq method [29] was utilized with a Kinnex 16S rRNA kit (Pacific Biosciences, Menlo Park, CA, USA). A total of 1–2 ng of DNA per sample was used for the amplification of full-length 16S genes (V1–V9 regions) with barcoded forward and reverse 16S primers from a Kinnex PCR 12-fold kit (Pacific Biosciences, Menlo Park, CA, USA). Following amplification, the pooled 16S PCR amplicons were purified using SMRTbell cleanup beads (Pacific Biosciences, Menlo Park, CA, USA). Kinnex adapters were then ligated to the ends of the barcoded full-length amplicons, and the PCR-amplified 16S fragments were treated with Kinnex enzyme and ligase to assemble the 16S segments into a linear array of approximately 19 kb, utilizing a Kinnex concatenation kit (Pacific Biosciences, Menlo Park, CA, USA). The final SMRTbell library was analyzed using an Agilent 2100 Bioanalyzer (Agilent Technologies, Santa Clara, CA, USA) to determine the size of the library fragments. Sequencing was conducted on a PacBio Revio instrument by Nextomics Biosciences Co., Ltd. (Wuhan, China).

### 2.4. Next-Generation Sequencing (NGS) of 16S rRNA Gene Amplicons

To investigate the changes of EFBs throughout the development of the *H. bachu* fruiting body (host *P. euphratica*), clean tissue samples were collected from the inner base of the fruiting body at three distinct developmental stages, including nascent, developing, and mature fruiting bodies, with three biological replicates for each stage. DNA was extracted using the TGuide S96 Magnetic Soil/Stool DNA Kit (Tiangen Biotech Co., Ltd., Beijing, China) following the manufacturer’s protocol. DNA concentration was measured with the Qubit dsDNA HS Assay Kit and Qubit 4.0 Fluorometer (Invitrogen, Thermo Fisher Scientific Inc., Waltham, MA, USA). To amplify the V3–V4 region of the 16S rRNA gene, a universal primer set (338F: 5′–ACTCCTAC–GGGAGGCAGCA–3′ and 806R: 5′–GGACTACHVGGGTWTCTAAT–3′) was employed. Sequencing was conducted on the Illumina NovaSeq 6000 platform (Illumina, Santiago, CA, USA) by Biomarker Technologies Co., Ltd. (Beijing, China).

### 2.5. Data Processing for Full-Length Sequencing and NGS of the16S rRNA Gene

Data processing was carried out with the support of BMK Cloud (Biomarker Technologies Co., Ltd., Beijing, China). The analytical pipeline included data quality control, species annotation, and analyses of alpha and beta diversity, along with assessments of differential species and functional characteristics. The QIIME2 feature classifier was employed to compare reads after quality control measures, which involved noise removal and the elimination of chimeras and mitochondrial sequences, using the Silva database (https://qiime2.org/). This alignment produced abundance tables of species composition across various taxonomic levels for each sample.

For all samples, alpha diversity indices—including observed species, Shannon, Simpson, Goods coverage, Chao1, ACE, and Pielou—along with beta diversity indices, were calculated using the R packages vegan, picante, and combinat. Differential analysis was conducted using LEfSe (https://huttenhower.sph.harvard.edu/galaxy/, accessed on 12 September 2024) [30] software to identify biomarkers across two or more groups. PICRUSt2 (https://github.com/picrust/picrust2, accessed on 16 September 2024) [31] was employed to predict the functions of representative sequences. Functional KO-pathway analysis was performed to forecast the KO results for different samples, and the corresponding functional pathway table was utilized to calculate abundance across various pathways. All boxplots were generated using the RandomcoloR (https://www.npmjs.com/package/randomcolor, accessed on 19 September 2024) and ggplot2 packages (https://ggplot2.tidyverse.org/, accessed on 10 September 2024).

### 2.6. Isolation and Identification EFBs from Helvella bachu Fruiting Bodies Using Culture Methods

The EFBs from *H. bachu* fruiting bodies associated with *P. euphratica* were isolated using four different media: Luria–Bertani (tryptone 10 g/L, Yeast extract 5 g/L, NaCl 10 g/L, pH 7); R2A (yeast extract powder 0.5 g, peptone 0.5 g, casein hydrolysate 0.5 g, glucose 0.5 g, soluble starch 0.5 g, dipotassium hydrogen phosphate 0.3 g, anhydrous magnesium sulfate 0.024 g, pyruvate 0.3 g, agar 15.0 g, pH 7.2 ± 0.2 [25]; TSA (cheese peptone 15.0 g, soybean papain hydrolysate 5.0 g, sodium chloride 5.0 g, agar 15.0 g, distilled water 1000 mL, pH 7.3 ± 0.2) [32], and YMG (glucose 4.0 g, yeast extract 4.0 g, malt extract 10.0 g, agar 15.0 g, distilled water 1000.0 mL, pH 7.2). The bacteria were purified and identified using universal primers 27F and 1492R.

## 3. Results

### 3.1. Discovery of EFBs in the Fruiting Body of Helvella bachu During Genome Sequencing Analysis

Clean tissue blocks were collected from the inner base of the *H. bachu* fruiting body (Figure 1A,B) to eliminate any risk of contamination. Genome sequencing was conducted using the MGISEQ-T7 and PacBio Sequel II platforms, yielding a total of 5.9 Gb of NGS reads and 7.2 Gb of PacBio HiFi reads, respectively. The generated reads were analyzed using Kraken 2.0, a taxonomic sequence classifier that assigns taxonomic labels to short DNA sequences [26].

The results indicated that 73% of the reads were fungal, while bacteria constituted 17% (Figure 1C). Subsequently, a BLAST search was performed using the 16S ribosomal RNA gene sequence of *E. coli* (MN900682.1) against the original sequenced data, with an e-value threshold set at 10^−5^, yielding 144 reads (Figure 1C). These reads were then queried against the NCBI bacterial database, also using an e-value threshold of 10^−5^. This analysis identified 27 bacterial genera, including *Acidovorax*, *Pedobacter*, *Agrobacterium*, *Stenotrophomonas*, *Aminobacter*, etc. (Appendix A). Consequently, it was suspected that EFBs may be present within the fruiting body of *H. bachu*.

### 3.2. QPCR Analyses Confirmed the Presence of EFBs in the Fruiting Body of Helvella bachu

Real-time quantitative PCR targeting the 16S rRNA gene was employed to detect bacteria in the clean tissues of the *H. bachu* fruiting body. The standard curve generated from the plasmid standard (Appendix A) and the melting curve (Appendix A) demonstrated the reliability of the method. The copy number of 16S rRNA in *H. bachu* tissue was quantified at 3.52 × 10^14^ copies/g, thereby confirming the presence of EFBs within the internal tissues of *H. bachu*.

### 3.3. Bacterial Abundance and Community Composition of Helvella bachu EFBs Associated with Three Types of Host Revealed by Full-Length 16S rRNA Gene Sequencing

#### 3.3.1. Bacterial Abundance and Community Composition

Clean tissues of *H. bachu* fruiting body associated with three hosts, *P. euphratica* (FPE), *P. pruinosa* (FPP), and *P. alba* var. *pyramidalis* (FPA) (Figure 2A, Table 1), were collected for full-length 16S rRNA gene sequencing using PacBio technology. After denoising and removing chimeric sequences, a total of 430,853 reads were obtained, yielding an average of 47,872 reads per sample (Appendix A).

The dilution curve (Appendix A) indicated that the sequencing data for all samples stabilized and reached saturation, suggesting that the sample data were both abundant and uniform, with FPA being particularly distinctive. Furthermore, the curve showed that saturation was achieved when the sequencing data reached 40,000 CCS (Appendix A), confirming the adequacy of the sequencing data.

At the amplicon sequence variant (ASV) level, a total of 632 features were identified across the three kinds of sample associated with different hosts. Among these, 121 ASVs were shared among all three samples, while 124 were unique to FPE, 87 to FPP, and 42 to FPA. Additionally, 95 ASVs were shared between FPE and FPP, 31 between FPE and FPA, and 132 between FPP and FPA (Figure 2B). In terms of abundance, the distribution of ASVs among the samples follows this order: FPE > FPP > FPA.

Species composition and abundance at various taxonomic levels were assessed for each sample. In total, 17 phyla, 32 classes, 88 orders, 135 families, 259 genera, and 367 species of EFBs associated with fruiting bodies of *H. bachu* from the three host plants were identified (Appendix A). Among these, Proteobacteria was the most abundant phylum found in the EFBs of all three host plants (Appendix A), followed by Bacteroidota. Together, these two phyla accounted for over 98% of the total reads across all samples. Additionally, Planctomycetota, Verrucomicrobiota, Patescibacteria, Acidobacteriota, Myxococcota, Actinobacteriota, and Firmicutes were also detected in the EFBs of the three host plants, albeit in lower proportions.

A total of 24 genera were identified with an abundance greater than 0.01% of the total (Appendix A). Notably, the genus *Stenotrophomonas* was abundant in the EFBs of all three host plants, accounting for over 20%. Other notable genera included *Variovorax*, *Acidovorax*, and *Pedobacter*, each contributing over 5% across all samples. There were some differences in abundance among the three host plants. The top three genera for FPE and FPP were *Stenotrophomonas*, *Variovorax*, and *Pedobacter*; however, the top three genera for FPA were *Variovorax* (29.26%), *Stenotrophomonas* (22.09%), and *Devosia* (17.49%).

A total of 367 species of EFBs were identified across three types of samples associated with different host plants. Among these, 98 species were shared among all three samples, while 51 species were unique to FPE, 58 to FPP, and 20 to FPA (Figure 2C). Additionally, 147 species were shared between FPE and FPP, 113 species between FPE and FPA, and 174 species between FPP and FPA. In terms of abundance, the distribution of species among the samples followed this order: FPP > FPE > FPA.

The common species shared among the three hosts included 34 species with an abundance greater than 0.01% of the total (Figure 2D and Appendix A). However, there were great differences among the three hosts. For FPE, six species had an abundance over 5%, listed in the following order: *Variovorax paradoxus* (29.26%), *Stenotrophomonas maltophilia* (13.72%), *Pedobacter panaciterrae* (12.25%), *Devosia riboflavina* (9.26%), *Devosia oryziradicis* (7.87%), *Stenotrophomonas rhizophila* (7.65%), and *Brevundimonas poindexterae* (5.21%). For FPP, six species also exceeded 5% abundance, listed as follows: *V. paradoxus* (22.91%), *Pedobacter steynii* (21.15%), *S. rhizophila* (14.47%), *S. maltophilia* (12.98%), *Acidovorax radicis* (11.81%), and *Bosea vestrisii* (9.19%). In the case of FPA, the leading species was *S. maltophilia* (41.48%), followed by *A. radicis* (20.89%) and *Agrobacterium radiobacter* (7.77%). Other notable species included *Acidovorax kalamii* (6.85%) and *V. paradoxus* (5.14%). In general, the species *S. maltophilia* and *V. paradoxus* were abundant in the fruiting bodies of *H. bachu* associated with all three hosts.

#### 3.3.2. Biomarkers for EFBs Associated with Various Host Plants

Biomarkers for various host plants were identified based on a linear discriminant analysis (LDA) value greater than 3 (Appendix A, Figure 2E). For FPE, the identified biomarker included Flavobacteriales, Weeksellaceae, and Oxalobacteraceae, along with specific species such as *P. steynii*, *Devosia limi*, and *Chryseobacterium vrystaatense*. In the case of FPP, key biomarker included Firmicutes, Gammaproteobacteria, and various members of *Pseudomonas*, such as *Pararhizobium*, *Rhizobium*, and several *unclassified Pseudomonas* species. FPA was characterized by the presence of *Devosia*, *Pseudoflavitalea*, and *Rhizobium sphaerophysae*, along with various uncultured bacteria.

#### 3.3.3. Diversity for EFBs in *Helvella bachu* Fruiting Bodies Associated with Various Host Plants

Various diversity indices, including the observed species index, Shannon index, Simpson index, Goods coverage index, Chao1 index, ACE index, and Pielou index, were calculated based on the abundance of ASVs (Appendix A). The alpha diversity index for FPE was the highest, followed by FPP, with no significant difference observed between the two groups (Figure 2F). In contrast, the α diversity index for FPA was the lowest, significantly lower than that of both FPE and FPP (*p* < 0.05).

The beta diversity of EFBs associated with three different host plants was analyzed using Principal Coordinates Analysis (PCoA). The percentage of variability explained by each principal component was as follows: PC1 accounted for 35.77%, while PC2 accounted for 23.08%. Ordinations based on the Bray–Curtis metric showed a clear separation among the EFBs associated with the three host plants (Figure 2G). Additionally, the bacterial communities in the FPA samples were more similar to those found in the FPP samples.

Analysis of similarities (ANOSIM) also revealed differences among the groups (Appendix A). However, the *p*-values for pairwise comparisons between the three host plants were greater than 0.05, indicating that these differences were not statistically significant. The R values for FPE, FPP, and FPA were 0.8519, 0.7407, and 0.4074, respectively. All R values fell within the range of 0 to 1, suggesting that the diversity of EFBs in different host plant groups was greater than that found within each individual host plant group.

#### 3.3.4. Functional Prediction and Functional Diversity Analysis

The functional KO-pathway analysis of the predicted KO results for the EFBs of *H. bachu* across all samples revealed that the enriched pathways were primarily associated with metabolism (Figure 3). Ranked by abundance from largest to smallest, these pathways included global and overview maps, xenobiotics biodegradation and metabolism, carbohydrate metabolism, amino acid metabolism, lipid metabolism, metabolism of cofactors and vitamins, biosynthesis of other secondary metabolites, energy metabolism, and metabolism of other amino acids. Additionally, pathways related to signal transduction within environmental information processing, endocrine systems of organismal systems, and transport and catabolism within cellular processes were also found to be enriched (Figure 3).

### 3.4. EFBs of Helvella bachu Fruiting Bodies at Different Developmental Stages

To investigate the dynamic changes in EFBs during the development of *H. bachu* fruiting bodies, samples from various developmental stages, including nascent (less than 3 cm in height, NF), developing (3–5 cm in height, DF), and mature (containing mature ascospores, MF) fruiting bodies (Figure 4A) were collected. These samples were analyzed using next-generation high-throughput sequencing of the 16S rRNA gene.

The dilution curves showed that all final profiles flattened, signifying that the sequencing data quantity was adequate (Figure 4B). The EFBs of *H. bachu* at various stages can be distinguished through PCA analysis following dimensionality reduction, accounting for 47.75% and 21.11% of the variance, respectively (Figure 4C). The bacterial operational taxonomic units (OTUs) were determined to be 1351, 1542, and 961 in the NF, DF, and MF, respectively (Appendix A). Only 66 OTUs were shared among all three stages (Figure 4D). Additionally, 104 OTUs were shared between the NF and DF samples, while 86 were shared between the NF and MF stages, and 103 were shared between the DF and MF. The number of unique OTUs at each stage was significantly greater than the number of common OTUs.

The EFBs across the three developmental stages included 30 phyla, 73 classes, 187 orders, 336 families, and 649 genera (Appendix A). At the phylum level, EFBs showed consistency across different stages, predominantly composed of Proteobacteria and Bacteroidetes, which together accounted for over 95% of the total. There was also a minor presence of Firmicutes and Actinobacteriota (Appendix A).

At the genus level, *Pedobacter* and *Acidovorax* were the most abundant genera shared among the three stages (Appendix A). At the NF stage, the abundances were as follows: *Stenotrophomonas* (15.36%), *Pedobacter* (15.04%), *Devosia* (14.32%), and *Acidovorax* (11.04%). At the DF stage, *Chryseobacterium* was the most abundant at 37.58%, followed by *Pedobacter* (18.93%), *Acidovorax* (6.00%), and an unclassified member of Xanthobacteraceae at 5.06%. At the MF stage, *Chryseobacterium* represented 27.91%, *Pedobacter* 22.49%, *Stenotrophomonas* 21.10%, *Acidovorax* 6.50%, and *Variovorax* 4.98%.

The ecological functions of the EFBs at three different developmental stages are illustrated for the top ten in functional abundance (Figure 4E). In general, consistency was observed across all three stages. Six ecological functions of the EFBs were highlighted across the samples, including chemoheterotrophy, aerobic chemoheterotrophy, nitrate reduction, and functions associated with animal parasitism or symbionts, nitrogen respiration, nitrate respiration, as well as those related to human pathogenic bacteria.

Differential analyses of functions of EFBs of *H. bachu* through pairwise comparisons at different developmental stages revealed that DNA replication, recombination, and repair at the NF stage were significantly more pronounced than at the DF stage (Figure 4F). Additionally, lipid transcription and metabolism in the NF stage were significantly higher than in the MF stage. In terms of aerobic chemoheterotrophy, the DF stage exhibited significantly greater levels than the MF.

### 3.5. EFBs from Helvella bachu Fruiting Bodies Using Culture Methods

The absence of bacterial colonies on the control plates indicated that surface contaminants were effectively removed, and the isolates were obtained from within the fungal tissue. A total of 46 bacterial strains were isolated and identified (Appendix A), belonging to four genera: *Pseudomonas*, *Stenotrophomonas*, *Chryseobacterium*, and *Variovorax*. Among these forty-six strains, thirty-four were classified under the genus *Pseudomonas* (73.91%), including seven strains of *Pseudomonas fluorescens*, four strains of *Pseudomonas gessardii*, three strains of *Pseudomonas azotoformans*, three strains of *Pseudomonas poae*, one strain of *Pseudomonas putida*, and one strain of *Pseudomonas simiae*. There were five strains of *Stenotrophomonas* (10.87%), comprising three strains of *S. maltophilia*, one strain of *S. rhizophila*, and one strain of *Stenotrophomonas* sp. Additionally, three strains of *Chryseobacterium* (6.52%), two strains of *Microbacterium*, one strain of *Serratia* sp., and one strain of *Variovorax boronicumulans* were identified.

## 4. Discussion

EFBs have emerged as a significant area of research within mycology, particularly for their roles in ecology, metabolism, and interactions with host fungi. *H. bachu* is a valuable ectomycorrhizal fungus associated with *P. euphratica*, a rare, ancient, and endangered species in desert riparian ecosystems. Despite numerous efforts, pure mycelial cultures have remained elusive, and attempts to cultivate its fruiting bodies have been unsuccessful for many years. This study confirmed the presence of EFBs within the fruiting bodies of *H. bachu* by a polyphasic approach, including genomic sequencing analysis, real-time quantitative PCR targeting the 16S rRNA gene, full-length sequencing and NGS of the 16S rRNA gene, as well as the culture methods. The genera *Stenotrophomonas*, *Variovorax*, *Acidovorax*, and *Pedobacter* were found to be abundant in *H. bachu* fruiting bodies associated with all three host plants, and these genera were also prevalent across different developmental stages. Notably, *S. maltophilia* and *V. paradoxus* were consistently detected by full-length 16S rRNA sequencing, and *S. maltophilia* was also isolated through culture methods. Functional KO-pathway analysis of the predicted KO results for EFBs in *H. bachu* revealed that metabolism-related pathways were predominantly enriched. This study not only confirmed the presence of EFBs in *H. bachu,* but also provided a comprehensive overview of their structure, functional potential, and dynamic changes throughout the maturation of *H. bachu* fruiting bodies, offering valuable insights that may facilitate the artificial domestication of this species.

### 4.1. EFBs Are Ubiquitously Present in Mushrooms, Especially in Ectomycorrhizal Mushrooms

EFBs resided within the vegetative or reproductive structures of fungi, with their presence documented in various species. Notable examples included saprophytic mushrooms such as *A. bisporus*, *Lentinus edodes*, *Pleurotus cornucopiae* [33], and Sang Huang [34], among others. However, EFBs are more commonly found in ectomycorrhizal fungi. These included Ascomycota species such as some species of *Morchella* [35,36] and *Tuber* [37,38,39], as well as Basidiomycota species, including *Tylopilus felleus*, *T. areolatus, Boletus queletii, Phlebopus portentosus* [19,40] *Tricholoma bicolor* [41], *Russula griseocarnosa* [42], *Amanita pantherina*, *Suillus placidus*, *T. felleus*, *Agaricus flocculosipes*, *Chlorophyllum molybdites* [43], and *F. luteovirens* [15]. In the present study, EFBs were confirmed in the fruiting bodies of *H. bachu* by a combination of methods, including genomic sequencing analysis, real-time quantitative PCR targeting the 16S rRNA gene, full-length sequencing and NGS of the 16S rRNA gene, as well as cultural methods. EFBs are ubiquitously present in mushrooms, particularly in ectomycorrhizal mushrooms.

### 4.2. Species Composition and Abundance of EFBs in Helvella bachu

Proteobacteria and Bacteroidetes were consistently identified as the predominant phyla in the EFBs of fruiting bodies associated with all three host plants (Appendix A) and across the three different stages (Appendix A). Together, these two phyla accounted for more than 95% of the total reads across all samples. It was reported that the relative abundance of Proteobacteria, Actinobacteria and Bacteroidetes were dominant in EFBs of *Cantharellales* and *Hydnum* [44].

At the genus level, some differences in abundance were observed among the EFBs in fruiting bodies associated with the three host plants (Appendix A). However, the genus *Stenotrophomonas* remained consistently abundant across all three host plants, accounting for over 20% of the total. Other notable genera, including *Variovorax*, *Acidovorax*, and *Pedobacter*, each contributed over 5% across all samples. These genera were also detected in the fruiting bodies at different stages, with *Pedobacter* and *Acidovorax* being the most abundant genera shared across all three stages. Different fungal groups harbor specific EFB communities. In ascomycetous mushrooms, these included *Bacillus*, *Mesorhizobium*, *Variovorax*, *Burkholderia*, *Pseudoma*, *Sphingobacteriaceae*, *Flavobacteriaceae*, *Ensifer*, *Rhizobium*, and *Rhodococcus* [38,45]. In ectomycorrhizal fungi, *Pedobacter*, *Pseudomonas*, *Variovorax*, *Chitinophaga*, *Ewingella*, and *Stenotrophomonas* were the dominant genera [22,46].

Full-length 16S rRNA gene sequencing, utilizing third-generation sequencing platforms such as PacBio SMRT, increases read lengths and confers species-level resolution [47,48]. In the present study, EFBs in the fruiting bodies associated with three hosts were identified by full-length 16S rRNA gene sequencing. Despite great differences among the three hosts, the species *S. maltophilia* and *V. paradoxus* were consistently abundant in the fruiting bodies of *H. bachu* associated with all three hosts. Other species, including *A. kalamii*, *A. radiobacter*, *B. poindexterae*, *B. vestrisii*, *D. oryziradicis*, *D. riboflavina*, *P. panaciterrae*, *P. steynii*, and *S. rhizophila*, were also identified, with varying abundances across the three hosts (Appendix A). Especially, *S. maltophilia* and *S. rhizophila* were also isolated through culture methods (Appendix A).

### 4.3. Potential Functions of EFBs in Helvella bachu

Endosymbioses have profoundly influenced the evolution of life and continue to shape the ecology of numerous species [10]. The widespread presence of EFBs in ectomycorrhizal mushrooms highlights their diverse and significant ecological roles. EFBs contribute to fungal health by serving as a bioenergy source, producing essential compounds like sugars, proteins, vitamins, and auxins [16,17,18], facilitating nutrient transport, enhancing respiration and environmental adaptability [19,20,21], acting as biological control agents against fungal diseases [11], and impacting the development, maturation, and aroma of mushrooms [22,23].

In this study, the functional KO-pathway analysis of the EFBs in the fruiting bodies associated with three host plants revealed that the pathways were predominantly enriched in metabolic processes, including primary, secondary, and energy metabolism (Figure 3). Primary metabolic pathways encompassed carbohydrate metabolism, amino acid metabolism, lipid metabolism, metabolism of cofactors and vitamins, and metabolism of other amino acids (Figure 3). These findings suggested that the EFBs may contribute to the growth of *H. bachu* by producing essential compounds such as sugars, proteins, and vitamins. Additionally, secondary metabolism and energy metabolism pathways were also identified as being significant.

The ecological function analysis of EFBs in *H. bachu* fruiting bodies at different developmental stages by APROTAX revealed that nitrogen respiration, nitrate reduction, and nitrate respiration were prominent across all developmental stages. This suggested that EFBs play a significant role in nitrogen cycling, providing nutrients for the host *H. bachu*. Other studies also have demonstrated that EFBs are crucial for carbon cycling [49], lipid metabolism [50], and nitrogen complex [51].

The species *S. maltophilia* and *V. paradoxus* were consistently the most abundant in the fruiting bodies of *H. bachu* associated with all three host plants. *S. maltophilia* has been reported to have several roles, including: (1) biological nitrogen fixation and secretion of plant hormones to promote plant growth [52]; (2) production of antibiotics and various enzymes to control pathogenic microorganisms and support plant growth [53,54]; and (3) enhancement of saline–alkaline resistance [55]. *A. radicis* has been shown to stimulate root growth and induce systemic resistance in plants [56], thereby benefiting agriculture by promoting plant growth and providing pest suppression under changing climatic conditions [57]. Future studies will further explore the functions of *S. maltophilia* and *V. paradoxus* in these aspects.

The dynamic changes of EFBs across the three developmental stages revealed higher nitrogen fixation in the NF and DF stages compared to the MF stage. In contrast, aerobic chemoheterotrophs were more prevalent in the MF stage than in the earlier two stages. Similarly, the *Allorhizobium–Neorhizobium–Pararhizobium–Rhizobium* complex plays a key role in nitrogen fixation, with this function diminishing as the fruiting bodies mature in *C. cibarius* [14]. These intricate interactions offer valuable insights for improving mushroom cultivation practices.

Lastly, species-specific symbiotic associations, such as those between *Lactarius deliciosus* and pine trees, or *Suillus luteus* and pine trees [58], are rare, typically occurring within certain taxa, including Boletales, Russulales, and Agaricales. Actually, some species of *Helvella* form associations with a broader range of host types, including conifers, *Castanopsis*, *Fagus*, *Populus*, and several species of *Quercus* [59]. While *H. bachu* may also establish symbiotic relationships with tree species other than *Populus* in the wild, such associations remain undocumented due to insufficient research and data. Future studies will be valuable in exploring both the host species and EFBs associated with *H. bachu*.

## 5. Conclusions

A polyphasic approach confirmed the presence of EFBs in the fruiting bodies of *H. bachu* and provided the first comprehensive overview of their structure, functional potential, and dynamic changes throughout fruiting body maturation. Future experiments could further validate the functions of these EFBs, which could be utilized to obtain pure cultured mycelia and incorporated into mycorrhizal synthesis for artificial propagation.

## Figures and Tables

**Figure 1 jof-10-00889-f001:**
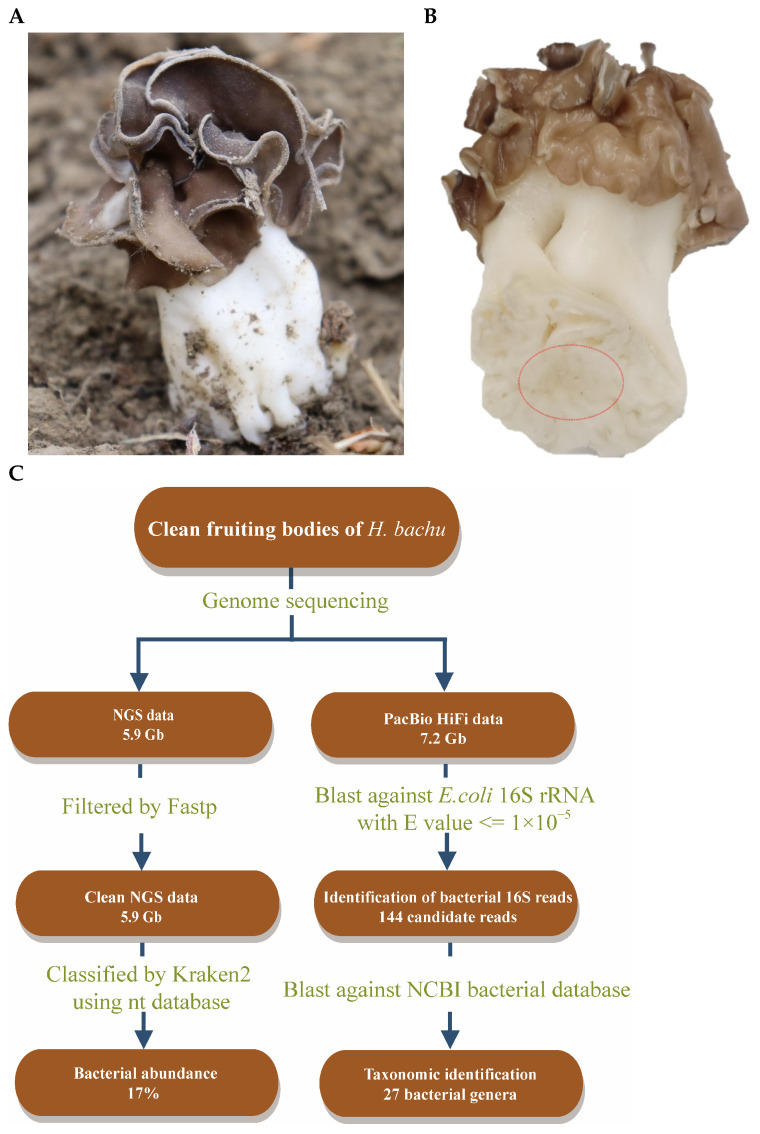
Discovery of EFBs in fruiting body of *Helvella bachu* during genome sequencing analysis. (**A**) Fruiting body of *H. bachu* in nature; (**B**) cleaned fruiting body of *H. bachu.* The sample location was marked with a red circle; (**C**) flowchart illustrating discovery of EFBs in fruiting body of *H. bachu*.

**Figure 2 jof-10-00889-f002:**
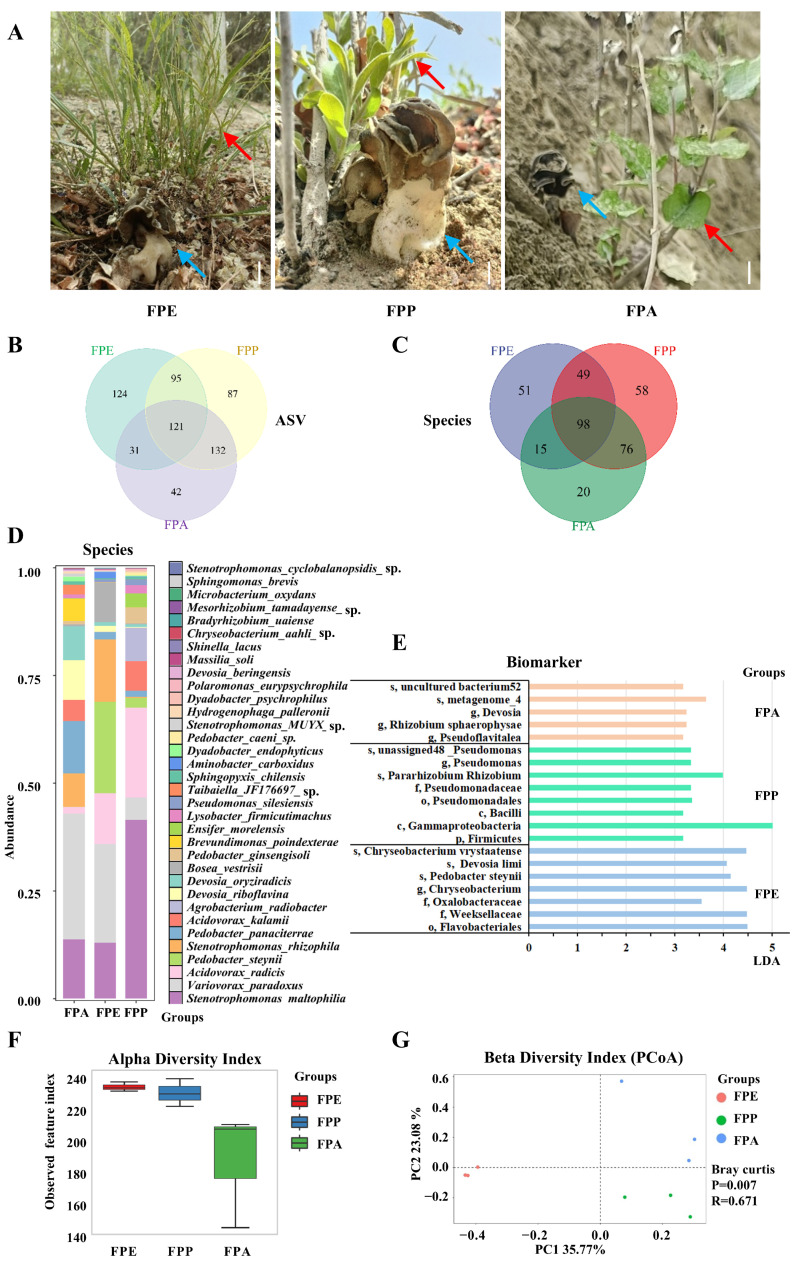
Abundance and community composition of *Helvella bachu* EFBs associated with different hosts revealed by full-length 16S rRNA gene sequencing. (**A**) Fruiting bodies of *H. bachu* associated with different hosts. Blue arrow is *H. bachu*, and red arrow is host plant. Bar: 1.5 cm; (**B**) Venn diagram of amplicon sequence variant (ASV) of *H. bachu* EFBs associated with three host plants; (**C**) Venn diagram of species of *H. bachu* EFBs associated with three host plants; (**D**) common species of *H. bachu* EFBs associated with three host plants; (**E**) biomarkers of *H. bachu* EFBs associated with three host plants; (**F**) alpha diversity index; (**G**) beta diversity index (PCoA). FPE: *H. bachu* fruiting body associated with *P. euphratica*; FPP: *H. bachu* fruiting body associated with *P. pruinosa*; FPA: *H. bachu* fruiting body associated with *P. alba* var. *Pyramidalis*.

**Figure 3 jof-10-00889-f003:**
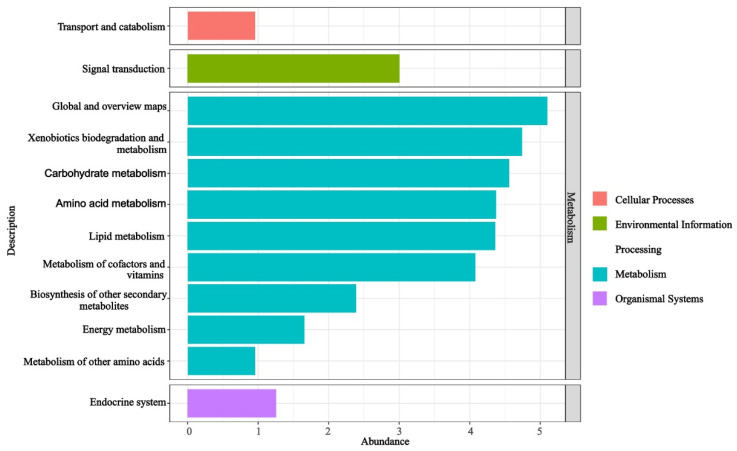
Functional pathways based on KO abundance of EFBs associated with three hosts.

**Figure 4 jof-10-00889-f004:**
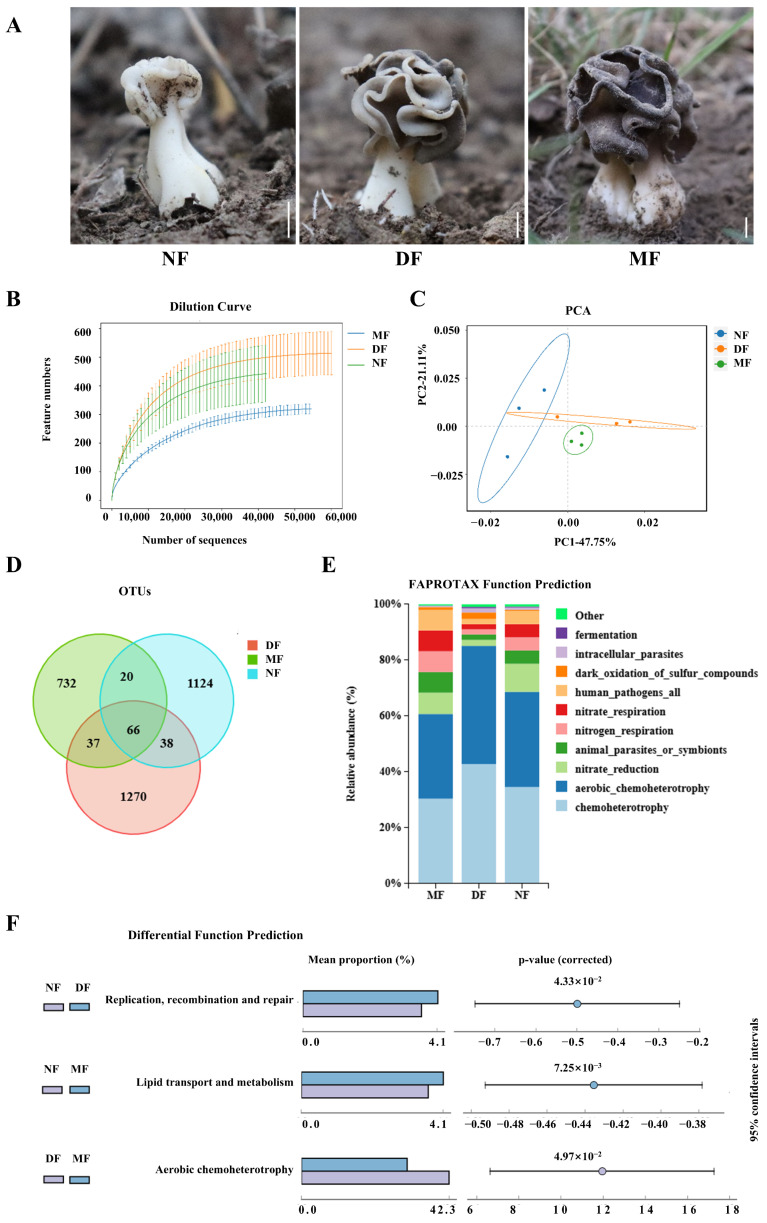
EFBs of *Helvella bachu* fruiting bodies at different developmental stages revealed by next-generation sequencing of 16S rRNA gene. (**A**) *H. bachu* fruiting body samples at different stages. Bar: 1cm; (**B**) dilution curve; (**C**) beta diversity index (PCA); (**D**) Venn diagram of OTUs of *H. bachu* EFBs at different stages; (**E**) ecological functions of EFBs of *H. bachu* at different stages; (**F**) differential analyses of functions of EFBs of *H. bachu* through pairwise comparisons at different stages. NF: nascent fruiting bodies (less than 3 cm in height); DF: developing fruiting bodies (3–5 cm in height); MF: mature (containing mature ascospores) fruiting bodies.

**Table 1 jof-10-00889-t001:** *Helvella bachu* samples used for full-length 16S rRNA gene sequencing.

Samples	Host Species	Address	Latitude	Longitude
FPE	*Populus euphratica*	Tarim University, Alar City, Xinjiang	N: 40°32′33.20″	E: 81°18′2.19″
FPP	*Populus pruinosa*	12 Regiment, Alar City, Xinjiang	N: 40°48′15.516″	E: 81°89′14.533″
FPA	*Populus alba* var. *pyramidalis*	11 Regiment, Alar City, Xinjiang	N: 40°35′1.79″	E: 81°41′26.49″

## Data Availability

The original contributions presented in this study are included in the article/Appendix A. Further inquiries can be directed to the corresponding authors.

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
