# Peer review of "Characterization of Endofungal Bacteria and Their Role in the Ectomycorrhizal Fungus Helvella bachu"

_jof, 2024, doi:10.3390/jof10120889_

Round 1
Reviewer 1 Report
I have no major comments, a few minor questions and remarks are added in the Detail comments section.
The paper is interesting and well-prepared, the subject, what is the interactions between ectomycorrhizal fungi and bacteria, especially MHB (Mycorrhiza helper bacteria) require further studies in numerous aspects. I noted one important issue, which is unclear or raise uncertainty.
In the lines 40-41, the Authors wrote:
<<Our previous studies have demonstrated that H. bachu is an ectomycorrhizal fungus, establishing a symbiotic relationship with poplar species such as Populus euphratica, Populus pruinosa, and Populus alba var. pyramidalis in desert riparian ecosystems.>>
This sentence is not incorrect. Nevertheless, the last, extensive study of ectomycorrhizal tree partners associated with Tuber species (also edible fungi) has shown surprisingly low specificity of Tuber species towards ECM tree partners. Not even a single Tuber species out of about 20 considered species was associated with only one genus of trees (Wilgan 2023, details for distinct ECM fungal species in the supplementary materials here: https://doi.org/10.3390/f14122407 ).
Both Tuber and Helvella form phylogenetic lineage /tuber-helvella (Tedersoo et al. 2010: http://dx.doi.org/10.1007/s00572-009-0274-x ), and exhibit similar environmental and soil preferences. Helvella species are associated, among others, with Quercus, Fagus, Pinus, Abies, Alnus, Castanea, and Castanopsis. Because Helvella is poorly studied and insufficiently recognized genus of ECM fungi (almost all species are represented by less than 10 sequences in UNITE database), the putative <exclusive preference> for Poplar species is probably the result of insufficient DNA data to proper estimate the species distribution and associated tree partners.
According to my knowledge, species-specific symbiotic associations, like between Lactarius delicious and pine trees only, or Suillus luteus and pine trees only (both are co-invasive in the southern hemisphere, e.g. in Australia and Chile (Wang et al. 2022: https://doi.org/10.3389/fmicb.2022.973483 ), are rare, and limited to three fungal orders - Russulales, Agaricales, and Boletales. These orders are younger in the evolution scale of time (Varga et al. 2019: http://doi.org/10.1038/s41559-019-0834-1 ) and far more diverse, than other orders of ECM fungi. In contrary, older orders of Basidiomycota (e.g. Thelephorales, Sebacinales, Cantharellales), and Ascomycota (Pezizales, including Tuber and Helvella), do not contain highly specialized ECM fungal species, which are associated with only one genus of trees.
Therefore, even if lines 41-42 are technically correct, they do not prove, that in the wild, H. bachu is associated with Populus trees only. Please, involve the explanation, which is given above, in the Introduction and Discussion section.
This comment concerns previous study (Wei, et al., Forests 2024, 15:721) more than the current one. Nevertheless, it's important issue. The <exclusive preference> of ECM fungi to the single genus of trees is rare phenomenon among deciduous trees, and involve only Boletales (eg. Leccinum, Alpova), Russulales (e.g. Lactarius), and Agaricales (e.g. Tricholoma, Cortinarius), but not Ascomycota, like Tuber and Helvella species.
Author Response
Comments 1:
The paper is interesting and well-prepared, the subject, what is the interactions between ectomycorrhizal fungi and bacteria, especially MHB (Mycorrhiza helper bacteria) require further studies in numerous aspects.
Response 1: Thank you very much for your positive comments. We have revised the manuscript in accordance with your suggestions and highlighted the changes in yellow.
Comments 2:
I noted one important issue, which is unclear or raise uncertainty. In the lines 40-41, the Authors wrote:
<<Our previous studies have demonstrated that H. bachu is an ectomycorrhizal fungus, establishing a symbiotic relationship with poplar species such as Populus euphratica, Populus pruinosa, and Populus alba var. pyramidalis in desert riparian ecosystems.>>
This sentence is not incorrect. Nevertheless, the last, extensive study of ectomycorrhizal tree partners associated with Tuber species (also edible fungi) has shown surprisingly low specificity of Tuber species towards ECM tree partners. Not even a single Tuber species out of about 20 considered species was associated with only one genus of trees (Wilgan 2023, details for distinct ECM fungal species in the supplementary materials here: https://doi.org/10.3390/f14122407 ).
Both Tuber and Helvella form phylogenetic lineage /tuber-helvella (Tedersoo et al. 2010: http://dx.doi.org/10.1007/s00572-009-0274-x ), and exhibit similar environmental and soil preferences. Helvella species are associated, among others, with Quercus, Fagus, Pinus, Abies, Alnus, Castanea, and Castanopsis. Because Helvella is poorly studied and insufficiently recognized genus of ECM fungi (almost all species are represented by less than 10 sequences in UNITE database), the putative <exclusive preference> for Poplar species is probably the result of insufficient DNA data to proper estimate the species distribution and associated tree partners.
According to my knowledge, species-specific symbiotic associations, like between Lactarius delicious and pine trees only, or Suillus luteus and pine trees only (both are co-invasive in the southern hemisphere, e.g. in Australia and Chile (Wang et al. 2022: https://doi.org/10.3389/fmicb.2022.973483 ), are rare, and limited to three fungal orders - Russulales, Agaricales, and Boletales. These orders are younger in the evolution scale of time (Varga et al. 2019: http://doi.org/10.1038/s41559-019-0834-1 ) and far more diverse, than other orders of ECM fungi. In contrary, older orders of Basidiomycota (e.g. Thelephorales, Sebacinales, Cantharellales), and Ascomycota (Pezizales, including Tuber and Helvella), do not contain highly specialized ECM fungal species, which are associated with only one genus of trees.
Therefore, even if lines 41-42 are technically correct, they do not prove, that in the wild, H. bachu is associated with Populus trees only. Please, involve the explanation, which is given above, in the Introduction and Discussion section.
This comment concerns previous study (Wei, et al., Forests 2024, 15:721) more than the current one. Nevertheless, it's important issue. The <exclusive preference> of ECM fungi to the single genus of trees is rare phenomenon among deciduous trees, and involve, but not Ascomycota, like Tuber and Helvella species.
Response 2: Thank you for providing us with valuable information and insightful suggestions. We agree that species-specific symbiotic associations are indeed rare, typically observed in certain taxa, such as Boletales (e.g., Leccinum, Alpova), Russulales (e.g., Lactarius), and Agaricales (e.g., Tricholoma, Cortinarius). Actually, some species of Helvella have been exhibited associations with a broader range of host species, including conifers, Castanopsis, Fagus, Populus, and several species of Quercus (Hwang et al., 2015). However, Helvella bachu has, to date, only been reported in the Xinjiang Uygur Autonomous Region of China, as well as parts of Pakistan and Iraq. It has been observed growing scattered or gregariously on sandy soil beneath Populus euphratica or P. bolleana. Our research in the H. bachu producing area of Xinjiang has exclusively identified Populus as its associated host. While H. bachu may form symbiotic relationships with other tree species in the wild except Populus, such associations have not yet been documented, likely due to a lack of comprehensive research and data.
We have incorporated the following contents in the Introduction and Discussion sections.
Lines 41-42: To date, H. bachu has been shown to establish a symbiotic relationship with various poplar species, including Populus euphratica, Populus pruinosa, and Populus alba var. pyramidalis.
Lines 511-518: Lastly, species-specific symbiotic associations, such as those between Lactarius deliciosus and pine trees, or Suillus luteus and pine trees (Wang et al., 2022), are rare, typically occurring within certain taxa, including Boletales, Russulales, and Agaricales. Actually, some species of Helvella form associations with a broader range of host types, including conifers, Castanopsis, Fagus, Populus, and several species of Quercus (Hwang et al., 2015). While H. bachu may also establish symbiotic relationships with tree species other than Populus in the wild, such associations remain undocumented due to insufficient research and data. Future studies will be valuable in exploring both the host species and EFBs associated with H. bachu.
References:
Hwang, J.; Zhao, Q.; Yang, Z.L.; Wang, Z.; Townsend, J.P. Solving the ecological puzzle of mycorrhizal associations using data from annotated collections and environmental samples–an example of saddle fungi. Environ Microbiol Rep. 2015, 7, 658–667.
Wang, R.; Wang, Y.L.; Guerin-Laguette, A.; Zhang, P.; Colinas, C.; Yu, F.Q. Factors influencing successful establishment of exotic Pinus radiata seedlings with co-introduced Lactarius deliciosus or local ectomycorrhizal fungal communities. Front. Microbiol. 2022, 13, 973483.
Reviewer 2 Report
The presented manuscript is devoted to a very important issue - the preservation of rare fungi in the ecosystem and their use as food.The endosymbiosis of fungi and bacteria is currently little studied, especially using high-precision methods of molecular biology. Solving environmental problems with the involvement of highly complex and highly dissolving methods of molecular biology allows us to obtain new important data and pose new promising questions
The manuscript is devoted to the definition of bacterial species, endosymbionts of higher fungi, which in turn are symbionts of several poplar species. As I have already reported earlier, the topic is quite modern and timely, since it allows us to more deeply delve into the various levels of the ecosystem, which includes plants-fungi-bacteria. This study does not repeat the existing ones, but brings new information to the understanding of interorganismal relationships at various levels, such as the participation of bacteria in optimizing the metabolism of fungi that form microises with poplar. An obvious advantage of the work can be considered modern molecular research methods: identification of bacterial species by rRNA and last-generation sequencing. The presented methodological protocols do not allow us to doubt the correctness of the results obtained. The conclusions made on the basis of these results correspond to the stated goal. As a non-critical remark, I recommend reducing the number of literary references: 30-40 are enough for an experimental article. Overall, the manuscript makes a favorable impression and is suitable for publication.I believe that the goal chosen in the manuscript is relevant, the methods for solving problems are very modern, and the results obtained are promising for further development of this topic.
Author Response
Comments:
Major comments
The presented manuscript is devoted to a very important issue - the preservation of rare fungi in the ecosystem and their use as food. The endosymbiosis of fungi and bacteria is currently little studied, especially using high-precision methods of molecular biology. Solving environmental problems with the involvement of highly complex and highly dissolving methods of molecular biology allows us to obtain new important data and pose new promising questions.
Detail comments
The manuscript is devoted to the definition of bacterial species, endosymbionts of higher fungi, which in turn are symbionts of several poplar species. As I have already reported earlier, the topic is quite modern and timely, since it allows us to more deeply delve into the various levels of the ecosystem, which includes plants-fungi-bacteria. This study does not repeat the existing ones, but brings new information to the understanding of interorganismal relationships at various levels, such as the participation of bacteria in optimizing the metabolism of fungi that form microises with poplar. An obvious advantage of the work can be considered modern molecular research methods: identification of bacterial species by rRNA and last-generation sequencing. The presented methodological protocols do not allow us to doubt the correctness of the results obtained. The conclusions made on the basis of these results correspond to the stated goal. As a non-critical remark, I recommend reducing the number of literary references: 30-40 are enough for an experimental article. Overall, the manuscript makes a favorable impression and is suitable for publication.
I believe that the goal chosen in the manuscript is relevant, the methods for solving problems are very modern, and the results obtained are promising for further development of this topic.
Response:
Thank you very much for your positive comments.
Following your suggestion, some literatures have been deleted.
Reviewer 3 Report
The manuscript Characterization of Endofungal Bacteria and Their Role in the Ectomycorrhizal Fungus Helvella bachu studies EFBs in the fruiting bodies of H. bachu during genome sequencing analysis. Their presence was firmed through fluorescent staining and real-time quantitative PCR targeting the 16S rRNA gene. The bacterial abundance and community composition of EFBs associated with three types of Populus spp. were then characterized using full-length 16S rRNA gene sequencing, along with predictions of their functional roles.
the EFBs isolated from H. bachu fruiting bodies using culture methods showed partial consistency with the high-throughput sequencing results. This study confirmed the presence of EFBs in H. bachu and provided a comprehensive overview of their structure, functional potential, and dynamic changes throughout the maturation of H. bachu fruiting bodies.
The manuscript is robust, featuring an excellent experimental design, employing cutting-edge techniques and yielding compelling results. It is a work that will be well-received by the readers of the Journal of Fungi.
-
Author Response
Comments:
The manuscript Characterization of Endofungal Bacteria and Their Role in the Ectomycorrhizal Fungus Helvella bachu studies EFBs in the fruiting bodies of H. bachu during genome sequencing analysis. Their presence was firmed through fluorescent staining and real-time quantitative PCR targeting the 16S rRNA gene. The bacterial abundance and community composition of EFBs associated with three types of Populus spp. were then characterized using full-length 16S rRNA gene sequencing, along with predictions of their functional roles.
The EFBs isolated from H. bachu fruiting bodies using culture methods showed partial consistency with the high-throughput sequencing results. This study confirmed the presence of EFBs in H. bachu and provided a comprehensive overview of their structure, functional potential, and dynamic changes throughout the maturation of H. bachu fruiting bodies.
The manuscript is robust, featuring an excellent experimental design, employing cutting-edge techniques and yielding compelling results. It is a work that will be well-received by the readers of the Journal of Fungi.
Response:
Thanks very much for your positive comments.